# Temporomandibular Disorders at the Preoperative Time of Orthognathic Surgery

**DOI:** 10.3390/diagnostics13182922

**Published:** 2023-09-12

**Authors:** Alice Vanzela Miotto, Danielle Veiga Bonotto, Jessica Sarha Cavalheiro Silva, Juliana Feltrin De Souza, Aline Monise Sebastiani, Rafaela Scariot

**Affiliations:** Department of Stomatology, School of Dentistry, Federal University of Paraná, 623 Prefeito Lothário Meissner Avenue, Curitiba 80210-170, PR, Brazil; alicevmiotto@gmail.com (A.V.M.); jessicasarahcavalheiro@hotmail.com (J.S.C.S.); julianafeltrin@ufpr.br (J.F.D.S.); rafaela_scariot@yahoo.com.br (R.S.)

**Keywords:** temporomandibular disorders, dentofacial deformity, sleep bruxism

## Abstract

Individuals seeking orthodontic treatment combined with orthognathic surgery (OS) have a high prevalence of temporomandibular disorders (TMDs), but the relationship between TMD diagnoses and dentofacial deformities (DFDs) is still controversial. Therefore, this cross-sectional study with a comparison group aimed to analyze the association between dentofacial deformities and TMDs. Methodology: Eighty patients undergoing OS were consecutively selected from the stomatology department of the Federal University of Paraná between July 2021 and July 2022. Forty patients who would undergo OS composed the group of participants with DFD, and forty who received other types of attention and did not present changes in the dental bone bases formed the group without DFDs (DFDs and no DFDs groups). The groups were matched for sex, age, and self-reported ethnicity. The diagnostic criteria for TMDs (DC/TMDs) were used to diagnose TMD based on the Axis I criteria. The psychosocial aspects, oral behaviors in wakefulness, and sleep bruxism were evaluated through the Axis II criteria. The data were analyzed with a 5% significance level. Results: The presence of DFDs was significantly associated with arthralgia (*p* = 0.01). The other types of TMDs were not associated with DFDs. Comorbidities, habits, and psychosocial variables were not associated with DFDs at a level of 0.05. (*p* > 0.05). In analyzing the participants with arthralgia, the ones with this condition presented higher frequencies of sleep bruxism (*p* = 0.046). Conclusions: Participants with DFDs presented a significantly higher frequency of arthralgia when compared to no DFDs ones. Sleep bruxism was associated with the occurrence of joint TMDs in these participants.

## 1. Introduction

Dentofacial deformities (DFD) correspond to dental malocclusion that is associated with skeletal pattern alterations [1]. Such deformities can be minimal, as in a slight projection of the chin, or extreme, as in a severe vertical maxillary excess or a hemifacial microsomia. The involvement may be in one or two bases of the bone in the vertical, horizontal, and transverse planes, both in isolation and in combination, thus causing different types of deformities. The main DFDs are skeletal class II, skeletal class III, skeletal biprotrusion, a skeletal anterior open bite, and a skeletal bilateral posterior crossbite [2].

The moderate and severe cases of DFDs require a combined treatment between orthodontics and orthognathic surgery (OS). Orthognathic surgery consists of repositioning the bone bases through osteotomies to align and correct the jaw’s position, thereby improving function and facial aesthetics. Pain management in orthognathic surgery is essential to enhance recovery, reduce hospital stay, and improve the whole experience of the patient [3]. The postoperative period of this surgery involves a difficult recovery process, with restrictions on food, difficulties in opening the mouth, significant swelling of the face, and frequent joint discomfort [4].

Among the reasons for performing OS, aesthetic purposes are frequently reported, as are functional improvements, including complaints related to temporomandibular dysfunction [5,6,7]. Although current concepts no longer consider occlusion to play a central role in the occurrence of TMDs, when it comes to dentofacial deformities, several studies [8,9] suggest a high prevalence of this condition in these individuals, especially in groups seeking surgical treatment. It is also important to note that many of these patients often have emotional problems, including anxiety and depression, due to the negative impact of the deformity [10,11].

Temporomandibular disorders (TMDs) encompass heterogeneous conditions involving the masticatory muscles, the temporomandibular joint (TMJ), or both, as well as their associated structures. Several studies have investigated TMDs in orthosurgical patients [12,13]; however, the way the deformity impacts specific TMD diagnoses still needs to be determined. There is controversy among the studies and a scarcity of papers that evaluate comparison groups using valid tools. The Research Diagnostic Criteria for Temporomandibular Disorders (RDC/TMD) [14] instrument followed the biopsychosocial model of TMD assessment and classification, which consists of Axis I (physical diagnoses) and Axis II (psychosocial aspects) criteria. It was updated in 2014 for the Diagnostic Criteria for TMDs (DC/TMDs) and recently validated for the Portuguese language in Brazil [15].

This study aimed to compare orthosurgical patients with dentofacial deformity (DFD group) and individuals without dentofacial deformity (no DFD group) regarding TMD diagnoses. 

## 2. Material and Methods

This is an observational cross-sectional study with a comparison group, which was developed in the Oral and Maxillofacial Surgery and Traumatology Service (OMSTS) facilities and the dental clinics of the Department of Stomatology at Federal University of Paraná (UFPR), Jardim Botânico campus, located in Curitiba in south of Brazil. The study went on for 36 months. Calibration within the examiners was performed using the kappa coefficient. To assess the inter-rater reliability, the Kappa coefficient was used (k:095/95% CI).

## 3. Ethical Aspects

The longitudinal study was approved by the Research Ethics Committee under protocol CAAE number: 52207821.9.0000.0102. In addition, the study followed all the recommendations of the Declaration of Helsinki [16] regarding research with humans. Individuals were invited to participate in the study and received information about the objectives and justifications for the research through an informed consent form (ICF). They received information about the benefits and risks to which they would be exposed. They were also aware that the treatment would continue, regardless of their refusal to participate in the research, and that they had the freedom to discontinue participation at any time. Individuals who consented to participate in the study signed the ICF and were included. Only the researchers had access to the questionnaires to ensure the confidentiality of the data.

## 4. Sample

The study included a total of 80 participants. The Open Epi Software, updated version 6 April 2013, calculated the sample size. The calculation was based on a previous study in which a prevalence of 31% was found in the general adult population [17]. For the group no DFDs, a prevalence of 63.8% was found in patients with DFDs who would undergo OS [12]. Thus, the calculation was performed with a bilateral confidence interval of 95% and a power of 80%, with a ratio of the no DFDs group to the DFDs group of 1:1. Among the 80 selected participants, 40 formed the group with DFDs, and 40 formed the group without DFDs.

The sample selection was performed as follows: for the group with DFDs, all individuals who would undergo OS by the OMSTS Service of UFPR and who met the inclusion criteria were invited to participate. The participants were recruited when they went through the clinical examination stage before OS. The inclusion criteria for the group with DFDs were the following: have dentofacial deformities not associated with cleft lip and palate or syndromes, requiring treatment through OS, over 18 years of age, accepted to participate in the research, and having signed the ICF. 

For each individual selected to the DFDs group, we looked for an individual matching the sex, age, and self-reported race in another dentistry clinic at the UFPR to maintain a more homogeneous sample and to eliminate potential biases. These individuals were approached in the first evaluation in other dental clinics of the UFPR. The individuals who agreed to participate and who met the inclusion and exclusion criteria were selected. The inclusion criteria for the group without DFDs comprised patients over 18 years of age, who were undergoing maintenance dental treatment, such as cleaning, bleaching, replacement of restorations, who signed the ICF, and who did not present dentofacial deformities. An experienced professional conducted a dentofacial deformity analysis.

The exclusion criteria for both groups were the presence of previous surgeries in the cervicofacial region, cognitive and neurological alterations, the diagnosis of arthritis and arthrosis in other joints, tooth pain, oral pathologies, and patients who used cyclobenzaprine in the week of the evaluation.

## 5. Data Collection

Demographic data were collected from all survey participants, such as sex, age, and self-reported race; they were divided into whites and nonwhites due to the n and low number of other races. In addition, the data were collected on drug use and the presence of comorbidities (fibromyalgia and arthritis/arthrosis), and data were collected on patients’ habits such as smoking (smoker and nonsmoker) and the amount of coffee intake (less than 3 cups and more than 3 cups per day).

For the analysis of DFD types, an experienced surgeon evaluated the participant’s profile and occlusion, as well as classified the face profiles into three categories: I, II, or III. Profile I is a straight profile, and profile II presents a negative step between the maxilla and mandible corresponding to patients with mandibular retrognathism. Profile III presents a positive step between the maxilla and mandible, thereby indicating anteroposterior maxillary deficiency, mandibular prognathism, or both [18]. They were also evaluated to consider mandibular asymmetry (higher than 4 mm) and vertical facial patterns: an anterior open bite, vertical maxillary excess, and vertical maxillary deficiency. 

All participants were diagnosed with TMDs through the DC/TMDs. The tool was used by trained and calibrated examiners. This tool consists of two axes: Axis I, which includes the information collected on the physical examination, and Axis II, which features emotional aspects. Both axes were used in this research [14].

Axis I consists of three questionnaires: TMD pain screening, DC/TMD symptom questionnaire, and TMD physical examination. To diagnose TMDs in patients, the DC/TMD procedures involved gathering the patient’s medical history, performing a physical exam that included muscle palpation (applying 1.0 kg pressure) and joint palpation (applying 0.5 kg pressure), assessing the presence of symptoms during mandibular function, and measuring the maximum opening of the jaw both with and without pain. Based on the collected findings, the tool identified two main categories of physical diagnoses (muscular and joint). It generated a decision flowchart and a table of diagnostic criteria to assist in the diagnostic process; painful TMD conditions in the muscles were classified as myalgia, local myalgia, myofascial pain, myofascial pain with spreading, and myofascial pain with referred pain; Axis I also classified headaches attributed to TMDs as present or absent (Figure 1). Regarding temporomandibular joint disorders, they can be structural or joint pain per se (arthralgia), and both diagnoses are performed independently on each side of the (TMJ. Structural disorders are classified as disc displacement with reduction, disc displacement with reduction and intermittent locking, disc displacement without reduction and without limitation of the opening, disc displacement without reduction and with limitation of the opening, and degenerative joint disease (Figure 2). As for arthralgia, it was considered present or absent with respect to joint pain. The degenerative joint disease was diagnosed through the symptom of crackling during mandibular function without complementary imaging.

Axis II incorporates behavioral instruments regarding pain, the psychological state, and psychosocial and behavioral functioning. For Axis II, the tools used were the Patient Health Questionnaire (PHQ-15), the Generalized Anxiety Disorder (GAD-7), the human body pain drawing to identify pain points in and beyond the face, and the Oral Behaviors Checklist (OBC), which was used to diagnose awake and sleep bruxism. The OBC is a self-evaluation tool with 21 questions in which the patient answers the weekly and monthly frequency of oral and parafunctional behaviors. According to the sum of the scores, the result is classified as no oral behaviors for scores up to four points, mild from four to twelve points, moderate from thirteen to nineteen points, and severe with more than twenty points. Based on the OBC data, combined with the clinical evaluation, the awake bruxism was classified as absent, infrequent, frequent, or very frequent according to the answers to the OBC questionnaire of the DC/TMDs. 

Sleep bruxism was classified as absent, possible, or probable. It was considered possible when it had a positive response in the OBC questionnaire and probable when, in addition to a positive response in the OBC, it presented at least one of the clinical signs: dental wear; marks on the soft tissue such as jugal mucus and tongue and/or muscle fatigue upon awakening; pain in the masseter and temporal muscle palpation; and/or masseter hypertrophy. [19] The PHQ-15 comprises 15 questions about physical symptoms such as nausea, dizziness, and body aches. The overall health conditions and the physical symptoms are classified according to severity level, from mild to severe. The instrument called GAD-7 consists of seven items to evaluate the symptoms of generalized anxiety disorder. The maximum total score is 21, where zero means the absence of anxiety, 5–9 means a mild degree, 10–14 indicates a moderate degree, and 15–21 means a severe degree of anxiety. Pain drawings are a self-applied tool represented by a drawing of the whole body and face in which the patient makes markings where they identify the pain. The score is the sum of the number of markings. A pilot study to verify the methodology and applicability of the questionnaires was conducted with 10 patients with DFDs who met the inclusion and exclusion criteria of the present study and who would undergo OS at the Oral and Maxillofacial Surgery and Traumatology Service (OMSTS) of the UFPR in Paraná, Brazil. There were no intercurrences during the pilot study, and, from that, the methodology was then implemented, and these ten patients were already included in the sample.

## 6. Data Analysis

The individuals were diagnosed according to the present or absence of the types of TMDs. Similarly, structural TMJ disorders were grouped as absent or present, thereby considering the sides of the TMJ (right and left) individually. Our study considered the presence of arthralgia, regardless of the affected side. As for the Axis II, the participants were classified as 1—without anxiety and mild anxiety—or 2—moderate and severe anxiety (GAD-7). Regarding the PHQ-15, the participants were classified as 1—without symptoms and mild symptoms—or 2—moderate and severe symptoms.

Regarding the OBC, parafunctional habits were dichotomized according to 1—absent and minor—or 2—present and very present. Wake bruxism was also dichotomized into 1—frequent and infrequent—or 2—frequent and very frequent.

The results obtained were submitted to descriptive and inferential statistical analysis. The categorical variables of the DC/TMDs between the groups were compared using the chi-square test or the Fisher exact test. For the numerical variables, the normality condition was evaluated by the Kolmogorov–Smirnov test. The comparison of numerical variables with non-normal distribution between groups was performed using the Mann–Whitney test. The variables of awake bruxism and sleep bruxism were compared with smoking and coffee habits in the general sample using the chi-square test or the Fisher exact test.

The level of significance adopted was 5%. The data were analyzed using the software *SPSS Statistics* v. 24 (IBM Corporation, Armonk, NY, USA).

## 7. Results

The sample was primarily composed of women, with 52 of them (65%) and 28 men (35%) being equally distributed between groups. The median age was 30 (19–61). Regarding the ethnicity, 65 participants (81.25%) were white, and 15 were nonwhite (18.75%). The groups were matched for sex (*p* = 1.0), age (*p* = 0.823), and self-reported race (*p* = 1.0). 

There was no association between the variables awake bruxism and sleep bruxism with the variables smoking and drinking more than three cups of coffee in the general sample (*p* > 0.05).

Table 1 shows that the groups’ presented homogeneity between some covariables, such as comorbidities and habits. 

Concerning the DFDs group, 7 participants presented the facial profile I, 12 had the facial profile II, and 21 had the facial profile III. A total of ten participants presented asymmetry. Regarding the vertical alteration patterns of the face, six participants presented vertical excess, six presented vertical deficiency, and nine presented an open bite. 

Table 2 shows the association between the diagnosis of TMDs and the groups. The presence of most TMD diagnoses was not associated with TMDs. However, individuals with DFDs presented a significantly higher frequency of arthralgia than those without DFDs (*p* = 0.01).

In analyzing the 18 participants who experienced joint pain (arthralgia) in the DFDs group, it was verified that 13 (72%) presented bilateral arthralgia, while 5 (28%) presented unilateral arthralgia. Also, 12 individuals (67%) presented disc displacement with reduction associated, with 4 of them presenting unilaterally, and in 8 (44.4%) of them it was present on both sides of the TMJ. In the group no DFDs, the four participants with arthralgia had no other joint disorders associated.

In terms of diagnosing disc displacement, our analysis found that, despite presenting similar results between the groups, 12 of them had painful symptoms related to their disc displacement, while 10 did not experience pain. Degenerative diseases in the TMJ were identified in one patient from the group with DFDs and in two from the group without DFDs. No patient had a disc displacement without reduction.

There was no difference in the maximum mouth opening between groups with and without DFDs. The median pain-free mouth opening value in the DFDs group was 43 mm (22–63 mm), and, in the group without DFDs, it was 45 mm (20–77 mm) (*p* = 1.0). The median mouth opening with pain was 48 mm (33–65 mm) in the DFDs group and 45 mm (30–78 mm) in the no DFDs group (*p* = 0.117).

Regarding the variables of Axis II of the DC/TMDs, there were no differences between the groups (*p* > 0.05) in anxiety levels (GAD-7), in physical symptoms (PHQ-15), and in pain in and beyond the face (Table 3). When considering the score for the pain points beyond the face on the pain drawing, there was also no difference between the groups; in the group with DFDs, the median pain score was 2 (0–19), and in the no DFDs group, it was 2 (0–12) (*p* = 0.262).

The variables of the subjects’ self-reported race, age, facial profile, and asymmetry variables were unrelated to arthralgia within the DFDs group (*p* > 0.05). Regarding sex, women had a higher prevalence of joint pain (*p* > 0.001). 

Table 4 shows the comparison between the participants with and without arthralgia (within the group with DFDs) in relation to the other variables of Axes I and II of the DC/TMDs and the parafunctional habits. Individuals with arthralgia presented a significantly higher frequency of sleep bruxism. The prevalence of probable sleep bruxism was 72% in individuals with joint pain compared to 59% in the group without joint pain (*p* = 0.046). 

## 8. Discussion

The diagnosis of TMDs is complex, mainly due to their different diagnostic methods and their multifactorial etiologies. Regarding the occlusal factor, after years of debate about the role of occlusal characteristics as causal or risk factors for TMDs, a low relevance for dental occlusion and the interarcaded relationship is currently attributed [13]. However, regarding DFDs, previous studies suggest a high prevalence of TMDs in this population, and a higher prevalence of pain and depression have also been reported compared to patients without DFDs [9,20].

This study focused on patients seeking OS, because it is necessarily understandable that temporomandibular dysfunctions are experienced in this population for the proper management and care of the patients before surgery. If patients undergo surgery with a previous pain, they will have more challenges in dealing with the postoperative period in functional and psychological terms. Also, long-lasting pain leads to changes in the central nervous system, thus causing central sensitization and increased postoperative pain sensitivity [21].

A recent systematic review showed that patients undergoing orthodontic surgical treatment have a higher incidence of TMDs when compared to a control population [9]. However, TMD subdiagnosis was not categorized in this systematic study, which only took into account the existence or absence of TMDs. Given the wide range of symptoms and treatment options available for TMDs, we do not believe that TMDs should be evaluated as a single diagnosis. Also, only two of the six studies that were a part of this systematic review used the RDC/TMD criteria to identify TMDs. Thus, to our knowledge, no previous studies have compared the diagnoses of DC/TMDs among patients with and without DFDs in the Brazilian population having used the DC/TMDs, which is the most accepted instrument for TMD diagnosis nowadays, thus emphasizing the importance of this study. 

The main finding in this study was that surgical patients with DFDs present a higher prevalence of arthralgia compared to a control population, thus corroborating another study [22]. Arthralgia is a type of TMD that is associated with peripheral etiological factors such as parafunction and joint overload (which can occur during sleep or wakefulness) [23]. The control of this condition encompasses a combination of noninvasive therapies, including patient education, self-care, cognitive behavioral therapy, physiotherapy, pharmacotherapy, and occlusal devices. When conservative therapies are not effective, minimally invasive therapies such as the intraarticular injection (IAI) of hyaluronic acid (HA) or corticosteroid (CS), arthrocentesis, or arthroscopy could be useful, but all these therapies should be associated with overload control [24].

The relationship between bruxism and the symptoms of temporomandibular disorders is deeply discussed in the literature, due to the complexity of the etiology and diagnosis of both conditions [25,26]. In the present study, a significant difference was found concerning probable sleep bruxism, which was more prevalent in the group with joint pain. Thus, we suggest that the joint overload caused by this condition contributes to the development of TMDs. However, the control of the sleep bruxism and consequently of the arthralgia could be very challenging during preoperative orthodontic preparation, which ends up making it impossible to use interocclusal devices. Thus, new treatment protocols should be investigated for these patients. 

In our study, we did not found any association between awake or sleep bruxism and smoking or drink coffee habits. However, is important observe that we had just seven smokers and three individuals that drank more than three cups of coffee in the general sample. So, it is possible that we did not find an association with these variables due to few individuals presenting these habits. According Bertazzo et al., more than eight small cups could be considered a risk factor for bruxism. Thus, we should not put away these habits as possible risk factors of bruxism [27].

Although there was no difference in the incidence of muscle TMDs between the groups, this study’s findings revealed that both groups had a high prevalence of the condition (55–60%). This could be associated with the period of high stress and poor sleep quality in the population over the data collection period, which was during the pandemic (2020–2021), when there was a significant increase in these symptoms reported by the population. The result of a higher prevalence may be due to the increased incidence of muscle TMDs occurring in the pandemic period compared to previous years [28], thus resulting in a higher prevalence than studies before this period [29,30].

It is also important to highlight that other studies analyzing specific subtypes of deformities found some different findings. For example, a systematic review suggested that class II skeletal profiles and hyper-divergent growth patterns were likely associated with an increased frequency of TMJ disc displacement and degenerative disorders. Another study that used cone beam tomography and the RDC/TMDs to diagnose TMDs found more prevalent bone changes in patients with the class II skeletal malocclusion [25]. Also, a study comparing the prevalence of TMDs in patients with dentofacial deformities associated with class III malocclusion found that it was similar to patients without dentofacial deformities [31]. Thus, it is necessary to consider a limitation of the present study in that the type of dentofacial deformity in each individual was not classified.

This study also investigated psychosocial variables between groups and their relationships with arthralgia. Although we found no associations in this study, this could also be due to having a small and restricted sample. Temporomandibular disorders are complex conditions, and their interrelationships should be seen through pain models, thereby inserting the biopsychosocial perspective in the evaluations. Thus, these variables should continue to be investigated in other studies.

Thus, we believe that the most important limitation of this study is the sample size (DFDs = 40 and no DFDs = 40), given that important associations could be found with a larger n. It was also impossible to segment the sample according to the type of deformity. Another constraint was that bruxism was identified without polysomnography, which made it impossible to deliver a definite diagnosis according to the international consensus on bruxism [19]. Also, while the clinic holds the authority for TMD diagnosis as per the DC/TMD guidelines, and the signs and symptoms show high accuracy for specific diagnoses, it is possible that the disc displacement without reduction (DDwoR) without a limited opening went undiagnosed in this study due to the absence of complementary exams, such as magnetic resonance imaging (MRI) of the TMJ.

Therefore, we suggest further prospective cohort studies with larger samples to better investigate our findings, thereby focusing on treatment options for these patients. We also recommend future studies to investigate additional factors that may contribute to TMDs in this population.

Finally, it is important to point out that understanding the profiles of these patients and their functional problems before OS can positively influence the outcome of ortho-surgical treatment, which is still a wildly neglected factor by surgeons. The prior identification of patients with TMDs, parafunction, or both, should imply a treatment plan mainly incorporating cognitive–behavioral approaches to help patients understand their need to maintain relaxed the masticatory muscles.

## 9. Conclusions

The prevalence of arthralgia was higher in orthosurgical patients with dentofacial deformity when compared to individuals without dentofacial deformity. Sleep bruxism was associated with the occurrence of joint TMDs in these patients. Detecting these conditions and carrying out adequate management before surgery can ensure a better prognosis. Thus, further studies should investigate the additional factors that may contribute to TMDs in this population and determine new protocols of preoperative management.

## Figures and Tables

**Figure 1 diagnostics-13-02922-f001:**
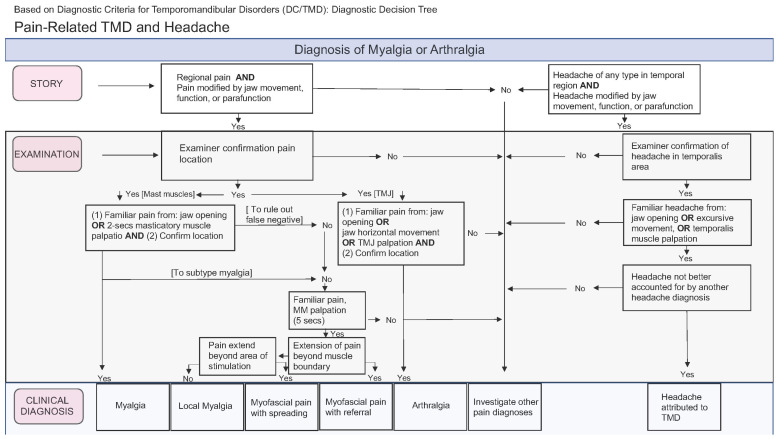
Flowchart of the criteria followed to myofascial pain, arthralgia and TMD Headache diagnose.

**Figure 2 diagnostics-13-02922-f002:**
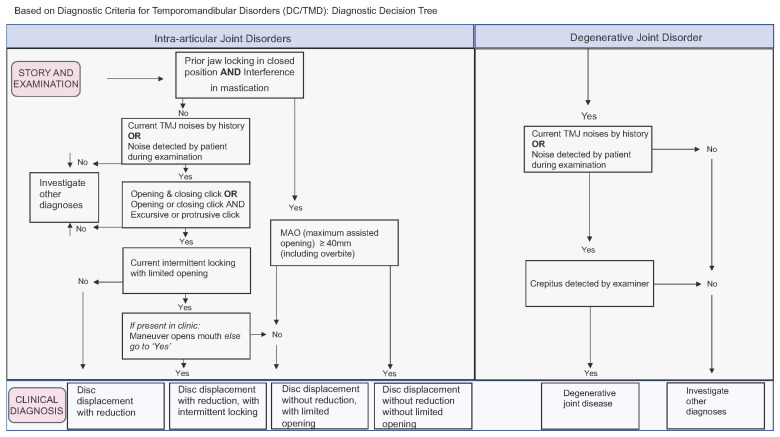
Flowchart of the criteria followed to temporomandibular joint disorders diagnose.

**Table 1 diagnostics-13-02922-t001:** The association of habits and comorbidities between the DFDs and no DFDs groups.

Variables		DFD *n* (%)	no-DFD *n* (%)	*p-Value*
Antidepressants Medication	No	37 (92.5)	36 (90.0)	1.000
	Yes	3 (7.5)	4 (10.0)	
Comorbidity	No	37 (92.5)	36 (90.0)	0.899
	Fibromyalgia	1 (2.5)	1 (2.5)	
	Gastroesophageal reflux	2 (5.0)	3 (7.5)	
Smokers	No	34 (85.0)	39 (97.5)	0.108
	Yes	6 (15.0)	1 (2.5)	
Coffee intake	0 to 3 cups	24 (60.0)	29 (72.5)	0.344
	>3 cups	16 (40.0)	11 (27.5)	

Note: Fisher exact test or chi-square test independent samples with a significance value of 0.05.

**Table 2 diagnostics-13-02922-t002:** Comparison of TMD diagnoses between DFDs and no DFDs groups.

Diagnoses		DFD *n* (%)	no-DFD *n* (%)	*p-Value*
Myalgia	Absent	18 (45.0)	16 (40.0)	0.821
	Present	22 (55.0)	24 (60.0)	
Left DDwR	Absent	27 (67.5)	28 (70.0)	1.000
	Disc displacement with reduction	13 (32.5)	12 (30.0)	
Right DDwR	Absent	20 (50.0)	29 (72.5)	0.066
	Disc displacement with reduction	20 (50.0)	11 (27.5)	
Arthralgia	Absent	22 (55.0)	36 (90.0)	**0.010**
	Present	18 (45.0)	4 (10.0)	
TMD Headache	Absent	30 (75.0)	29 (72.5)	1.000
	Present	10 (25.0)	11 (27.5)	

Note: Chi-square test of independent samples with a significance value of 0.05. Bold values indicate statistical significance. Acronyms: DDwR—disc displacement with reduction.

**Table 3 diagnostics-13-02922-t003:** Association between the Axis II DC/TMD variables between the groups.

Variables		DFD *n* (%)	no-DFD *n* (%)	*p-Value*
Physical symptoms	Absent and mild	15 (37.5)	19 (47.5)	0.498
	Moderate and severe	25 (62.5)	21 (52.5)	
Anxiety	Absent and mild	22 (55.0)	29 (72.5)	0.162
	Moderate and severe	18 (45.0)	11 (27.5)	
Pain in the face	Absent	15 (37.5)	19 (47.5)	0.176
	Present	25 (62.5)	19 (47.5)	
Pain Beyond the Face	Absent	16 (40.0)	21 (47.5)	0.370
	Present	24 (60.0)	19 (47.5)	

Note: Chi-square test of independent samples with a significance value of 0.05.

**Table 4 diagnostics-13-02922-t004:** Comparison of DC/TMD variables between participants with and without joint pain within the DFDs group.

Variables		No Arthralgia (22) *n* (%)	With Arthralgia (18) *n* (%)	*p-Value*
Myalgia	No	13 (59.0)	5 (27.0)	0.062
	Yes	9 (41.0)	13 (72.0)	
Articular Disorder Left	No DDwR	18 (81.0)	9 (50.0)	0.341
	With DDwR	4 (18.0)	9 (50.0)	
Articular Disorder Right	No DDwR	13 (59.0)	7 (38.0)	0.046
	With DDwR	9 (41.0)	11 (61.0)	
Pain Beyond the Face	No	9 (41.0)	7 (38.0)	1000
	Yes	13 (59.0)	11 (61.0)	
Physical symptoms	Absent and mild	11 (50.0)	4 (22)	0.104
	Moderate and severe	11 (50.0)	14 (77)	
Anxiety	Absent and mild	14 (63.0)	8 (44)	0.225
	Moderateand severe	8 (36.0)	10 (55.5)	
OBC	None/light	5 (22.0)	3 (16)	0.339
	Moderate–severe	17 (77.0)	15 (83)	
Sleep Bruxism	Absent	3 (13%)	5 (27)	**0.046**
	Possible	6 (27%)	0 (0.0)	
	Probable	13 (59%)	13 (72)	
Wake Bruxism	Absent; uncommon	13 (59%)	8 (44)	0.525
	Common; very common	9 (41%)	10 (55.5)	
	Yes	4 (18%)	2 (11)	
Coffee intake	0 to 3 cups	13 (59%)	11 (61)	1.00
	>3 cups	9 (40%)	7 (38)	

Note: Fisher exact test and chi-square test of independent samples with a significance value of 0.05. Bold values indicate statistical significance.

## Data Availability

The data presented in this study are available on request from the corresponding author. The data are not publicly available due to ethical restrictions.

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
