# Peer review of "Temporomandibular Disorders at the Preoperative Time of Orthognathic Surgery"

_diagnostics, 2023, doi:10.3390/diagnostics13182922_

Round 1

Reviewer 1 Report

Authors deal with the issue of temporomandibular disorders at the preoperative time of 2 orthognathic surgery. The topic is not new, but it is very important, as it basically determines the success of the operation and the quality of life after the operation. Many TMJ changes already exist before orthognathic surgery, other complaints only appear after the surgery.

According to their observations, arthralgia showed a significant correlation with the presence of DFD, while other parameters (for example, myalgia, headache, etc.) did not show a correlation.

The study was carefully planned, and the selection served to achieve the set goals. The examined clinical parameters and the applied statistical methods were relevant in both patient groups.

As a result of a well-planned study, the authors made important findings and justified exclusions.

The work is original and important, contributing to a better understanding of the relationships between DFD, TMD and other clinical symptoms.

I support its publication without changes.

Reviewer 2 Report

Re: diagnostics-2514569

Temporomandibular disorders at the preoperative time of orthognathic surgery

The relationship between dentofacial deformity (DFD) and temporomandibular joint disorder (TMD) is still controversial, and no conclusion has been reached yet. The present study compared the TMJ symptoms between patients with and without DFD. Thus, this study seemed to have some meaning in the field of orthognathic surgery. However, there are some unclear points which are needed to be improved.

1.       Is sample with non-DFD a volunteer? There is no clear description. Was the presence of TMD symptoms such as pain taken into consideration when recruiting non-DFD? 

2.       Why smoking and taking coffee were included in the evaluation factors?

3.       How was the disc displacement assessed? MRI is necessary for accurate assessment. 

4.       If the asymmetry cases were included in this study, the TMD should be evaluated at the deviation/non-deviation sides, not right/left sides.

Reviewer 3 Report

**Introduction:**

- The introduction provides a concise overview of dentofacial deformities (DFD) and their association with dental malocclusion and skeletal alterations.

- However, it might be helpful to include a brief explanation of what orthognathic surgery (OS) is, especially for readers who are not familiar with the term.

**Content Organization:**

- The article covers a range of topics related to dentofacial deformities, including their types, reasons for performing orthognathic surgery, the connection to temporomandibular disorders (TMD), ethical aspects, sample selection, data collection methods, and data analysis.

- While the article covers various aspects, the flow of content seems a bit disorganized. Consider restructuring the content to create a more coherent narrative flow.

**Research Methodology:**

- The article outlines the observational cross-sectional study design and the matching of groups based on sex, age, and self-reported race.

- The rationale for sample size calculation is provided, which is important for research transparency.

- However, it would be helpful to include more information about the recruitment process, potential biases, and the generalizability of the results.

**Data Collection:**

- The article describes data collection methods, including demographic information, diagnostic criteria for TMD (DC/TMD), and additional questionnaires to assess psychological factors.

- The inclusion of information about the pilot study helps validate the methodology used.

**Results:**

- The results are presented in terms of participant characteristics, TMD diagnoses, prevalence of arthralgia, and associations with sleep bruxism.

**Discussion:**

- The discussion delves into the findings related to the prevalence of TMD and arthralgia among ortho-surgical patients with dentofacial deformities.

- However, the discussion seems a bit repetitive in some sections, and it might benefit from a clearer structure to address each finding and its implications.

**Conclusion:**

- The conclusion summarizes the main findings of the study regarding the prevalence of arthralgia and its association with sleep bruxism.

- It could be expanded to discuss the potential clinical implications and future research directions.

**Overall Considerations:**

- The article covers an important area of research and provides insights into the relationship between dentofacial deformities, TMD, and related factors.

- However, the article would benefit from clearer organization and a more concise writing style.

- Consider breaking down some of the lengthy paragraphs into smaller, focused sections for improved readability.

- In the conclusion, highlight the clinical implications and potential areas for further research.

Round 2

Reviewer 2 Report

Re: diagnostivs-2514569-revised version

Temporomandibular disorders at the preoperative time of orthognathic surgery

The revised version was much better than the original version.

1. One more point should be reconsidered before reaching the acceptable level in this journal.

Smoking and taking coffee factors were omit in the revised version. However, the hypothesis that these factors could be related with the bruxism was very important point. Thus, the data should be inserted into the text and showed the conclusion “smoking and taking coffee was not related with bruxism” with hypothesis and the previous reports which described the relationship between smoking, taking coffee and bruxism. And the reason should be discussed the reason why these factors depended to the bruxism and TMD.

2. The TMD should be evaluated at the deviation/non-deviation sides. This is very important to decide the relationship between facial deformity with asymmetry and TMD. The response said no data for it. However, cephalogram should be taken for all patients before and after surgery.

Round 3

Reviewer 2 Report

Now, the revised manuscript was much bettter than the original version and I think this can achieve the level of the journal.